# Maize Streak Virus: Single and Gemini Capsid Architecture

**DOI:** 10.3390/v16121861

**Published:** 2024-11-29

**Authors:** Antonette Bennett, Joshua A. Hull, Mario Mietzsch, Nilakshee Bhattacharya, Paul Chipman, Robert McKenna

**Affiliations:** 1Department of Biochemistry and Molecular Biology, College of Medicine Center for Structural Biology, McKnight Brain Institute, University of Florida, Gainesville, FL 32610-0245, USA; mario.mietzsch@ufl.edu (M.M.); joshua.hull@duke.edu (J.A.H.); pchipman@ufl.edu (P.C.); 2Biological Science Imaging Facility (BSIR), Department of Biology, Florida State University, 89 Chieftain Way, Tallahassee, FL 32306-4370, USA; nilakshee.bhattacharya@duke.edu

**Keywords:** cryo-EM, geminivirus, icosahedral, capsid, plant, pathogen

## Abstract

*Geminiviridae* are ssDNA plant viruses whose control has both economical and agricultural importance. Their capsids assemble into two distinct architectural forms: (i) a T = 1 icosahedral and (ii) a unique twinned quasi-isometric capsid. Described here are the high-resolution structures of both forms of the maize streak virus using cryo-EM. A comparison of these two forms provides details of the coat protein (CP) and CP–CP and CP–genome interactions that govern the assembly of the architecture of the capsids. Comparative analysis of other representative members of *Geminiviridae* reveals structural conservation of 60–95% compared to a sequence similarity of 21–30%. This study provides a structural atlas of these plant pathogens and suggests possible antiviral-targetable regions of these capsids.

## 1. Introduction

Viruses are a major source of plant pathogens and cause economic losses of approximately USD 30 billion globally each year [1]. *Geminiviridae* are a family of pathogenic viruses that cause economic havoc and loss of important agricultural plants worldwide, including tomato, maize, barley, sugar cane, wheat, grape, and others (reviewed in [2]), and have been designated a “recent emerging virus of the world” [1,3,4]. They are transmitted by a wide variety of insect vectors, such as whiteflies, leafhoppers, treehoppers, and aphids (reviewed in [5]). The infected plants typically exhibit abnormal leaf coloration and stunted growth, which result in significant loss in crop yields (as reviewed in [6,7,8,9,10]). The *Geminiviridae* are subdivided into fourteen genera—*Mastrevirus*, *Curtovirus*, *Capulovirus*, *Begomovirus*, *Becurtovirus*, *Eragrovirus*, *Grablovirus*, *Turncurtovirus*, *Maldovirus*, *Opunvirus*, *Citildovirus*, *Mulcrilevirus*, *Topilevirus*, and *Topocuvirus*—based on genome organization, host range, and insect vector [5]. One of the most extensively studied genera is the *Mastrevirus*, which has a monopartite genome and is transmitted by the insect vector *Cicadulina mbila* [5,7,11]. Maize streak virus (MSV) has been ascribed as a prototypic member of the genus, and the Nigerian strain (MSV-A[NG1]) causes significant loss of maize, wheat, oats, barely, rye, millet, sorghum, sugarcane, and napier fodder crops (reviewed in [6,8]). This loss in crops is detrimental to subsistence farmers in sub-Saharan Africa and India [7] and is reported to cause damages of USD 120–480 million per year [8]. Currently, various methods aimed at geminivirus control include but are not limited to insecticides to combat the insect vector, transcriptional gene silencing, CRISPR/cas, defective interfering DNA, and replication protein (Rep) and CP antibodies (reviewed in [12]). Several of these antiviral strategies require detailed knowledge of the virus protein–protein, virus protein–genome, and virus–host interactions.

The infectious MSV-A[NG1] geminivirus capsid packages a 2.7 kb circular ssDNA genome, while a defective T = 1 single capsid has been shown to package subgenomic segmented ssDNA, 0.2–1.6 kb [13]. The geminivirus genome contains a stem-loop motif in the large intergenic region (LIR), which is conserved in all geminiviruses and required for genome replication. In addition, the genome is composed of four ORFs, which are bidirectionally transcribed and translated to produce the coat protein (CP), replication proteins (Rep and RepA), and a movement protein (MP) [14,15,16]. The CP is multifunctional, accumulates in the nucleus, interacts with the genomic DNA, transports the DNA from the nucleus to the cytoplasm, assembles the unique virus capsid, and is required for the encapsulation of ssDNA, insect transmission, and systemic infection [15,16]. The geminivirus capsid architecture is unique among all virus families [5], with a twinned quasi-isometric incomplete T = 1 capsid (geminate) structure [17,18]. Currently, there is one near-atomic and one high-resolution structure of *Geminiviridae* and *Begomovirus* genus determined: African cassava mosaic virus (ACMV) (PDB ID:6EK5) and Ageratum yellow vein virus (AYVV) (PDB ID:6F2S) at resolutions of 4.2 Å and 3.3 Å, respectively [19,20]. The authors of the AYVV structures identified two unique CP monomers at the equatorial twofold axis (interface between the two capsid heads) [19], with an extended N-terminal domain and hexanucleotide that were suggested to be the conserved stem loop observed in all geminiviruses and were both proposed to be essential for the unique gemini capsid assembly [19].

Although *Geminiviridae* contains 14 genera, high- and near-subatomic resolution structures are available only for the Begomovirus genus [19,21]. In order to effectively correlate the capsid structure of the genera with specific aspects of the virus life cycle of this pathogenic, agriculturally important virus, we determined the structure of the MSV-A[NG1] gemini capsid and the non-infectious, single T = 1 icosahedral particle to 3.17 and 3.72 Å resolution, respectively. These structures are the first high-resolution structures of a Mastrevirus and the first T = 1 geminivirus structure that have been determined to date. Both the Mastrevirus and Begomovirus structures and the predictive power of AlphaFold were used to generate representative CP models for all 14 genera. Additionally, phylogenic analysis of representative members from each genus was used to select two highly divergent CP models to generate a sequence and structure alignment. Structural annotation and mapping of the geminivirus capsid reveal regions of the capsid that are highly conserved and are the determinant of the secondary structures of the CP, which is important for the assembly of the geminivirus capsid. Additionally, there are variable regions (VRs) of the capsid among the different genera that most likely are determinants of vector and host specificity, which also play a significant role in vector transmission. These regions can be potential “druggable” targets for vector transmission of these pathogenic viruses.

## 2. Materials and Methods

### 2.1. Production and Purification

MSV-A[NG1] gemini and single capsids were produced by agroinoculation of maize leaves [22,23]. The symptomatic leaves were harvested ~25 days post inoculation, sliced into small pieces (~3 cm in length), and rapidly frozen with liquid nitrogen. The purification of the capsids from the frozen leaves is described in detail in [24]. In brief, the frozen, crushed leaves were blended with a mixture of buffer A (0.1 M sodium acetate (NaAc) pH 5.2, 1 mM ascorbic acid, and 1 mM EDTA), cellulase, and hemicellulase. The soluble (virus-containing) fraction was separated from unblended plant material by chloroform extraction. The aqueous layer containing the soluble virus was collected and PEG-precipitated with 10% PEG 8000 and 125 mM NaCl and by stirring overnight at 4 °C. The PEG-precipitated virus was collected by pelleting at 10,000× *g* for 90 min. The PEG pellet was resuspended in buffer B (0.1 M NaAc pH 4.8). The sample was loaded onto a 5 mL 15% sucrose cushion and centrifuged at 208,000× *g* for 3 h. The sucrose cushion pellet was resuspended in 1 mL buffer B, loaded onto a 5–40% step sucrose gradient, and centrifuged at 24,100× *g* for 3 h. The visible blue virus-containing fractions under white light were collected and dialyzed against buffer B and concentrations determined based on nucleic acid content, with an OD at 260 nm and an extinction coefficient of 7.5 for further characterization.

### 2.2. Verification of Virus Purity and Integrity

#### 2.2.1. Negative Stain Electron Microscopy (EM)

Aliquots (5 µL) of each purified MSV-A[NG1] sample at 1 mg/mL were loaded onto carbon coated copper EM grids (Electron Microscopy Sciences, Hatfield, PA, USA, cat# CF400-CU) for 2 min and negatively stained with 5 µL of 2% uranyl acetate for 20 s. The grids were air-dried and examined with a JOEL 1200 EX transmission electron microscope. The instrument was set to collect images at 50,000× magnification and on film.

#### 2.2.2. SDS-PAGE

Aliquots (10 µg) of MSV-A[NG1] samples were denatured by boiling for 5 min in the presence of 1–2% *v*/*v* 2-mercaptoethanol and loaded onto a 12% SDS-PAGE gel. The gel was stained with silver stain (Thermo Fisher, Waltham, MA, USA) and visualized on the GelDoc system (Bio Rad, Hercules, CA, USA).

### 2.3. Cryo-EM

#### 2.3.1. Data Collection

Three microliters of MSV-A[NG1] at ~1.0 mg/mL were applied to glow discharged, C-flat holey carbon grids (Protochips, Inc., Morrisville, NC, USA)) and vitrified with a Vitrobot Mark IV (FEI Co., Ltd., Waltham, MA, USA). The grids were screened with a Tecnai G2 F20-TWIN transmission electron microscope operated at 200 kV under low-dose conditions (~20 e^−^/Å^2^) for suitable particle distribution and ice thickness prior to high-resolution data collection. Cryo-electron micrograph movie frames were collected on a Titan Krios electron microscope (Thermo Fisher, Waltham, MA, USA) operated at 300 kV with a K2 detector (Gatan, Pleasanton, CA, USA) at the Southeastern Center for Microscopy of MacroMolecular Machines (SECM4). Micrograph frame alignment was performed using the MotionCor2 application with dose weighting [25]. The data collection parameters are summarized in Table 1.

#### 2.3.2. Single-Particle Reconstruction

The cisTEM software package was used to determine the three-dimensional image of both MSV-A[NG1] gemini and single capsid [26]. CTFFIND4 [27], a subroutine in cisTEM, was used to determine the defocus range and contrast transfer function (CTF) estimate per micrograph and eliminate poor-quality micrographs. cisTEM’s autopicking function was used to determine the particle position and to extract the selected particles for the next step of processing [28]. Two-dimensional classification of the selected particles was used to remove damaged particles as well as provide input for ab initio three-dimensional model generation [29]. A de novo model was generated by modifying cisTEM’s ab initio 3D default settings to impose D5 symmetry for gemini capsids and I symmetry for the single icosahedral capsid [30]. The initial model was auto-refined, and the final density map was sharpened with a pre-cutoff B-factor value of −90 Å^2^ and variable post-cutoff B-factors of 0, 20, and 50 Å^2^. The sharpened maps were used for both side and main chain assignments. The process is outlined in Appendix A.

#### 2.3.3. Model Building

A model was generated by AlphaFold based on the amino acid sequence. A 60-mer oligomer was constructed using the oligomer generator in ViperDB and fitted into the density map using the Chimera “fit in map” option for the single capsid. In order to maximize the correlation coefficient, the voxel size was adjusted and the model refitted. The EMAN2 subroutine e2proc3d.py was used to resize the map to the appropriate voxel size and to convert to CCP4 format using the program MAPMAN. The program Coot was used to manually refine the main- and side-chain models. The model was further refined using the program PHENIX, and the final statistics are reported in Table 1. The gemini capsid, however, had an additional element of model refinement in order to improve the resolution of the map. After the initial atomic model was built, a model map was generated using EMAN2’s e2pdb2mrc.py command. From this, RELION3 was used to generate a mask using relion_mask_create [31]. The mask was inverted as necessary to maintain the handedness of the uncorrected cisTEM map using Chimera’s vop zFlip followed by vop resample onGrid commands [32]. This mask was used in cisTEM for further local refinement and sharpening of the final map.

An initial MSV-A[NG1] model was generated using SWISS-model [33] using the deposited AYVV structure (6f2s) [19] as a template. This initial model was docked into the MSV-A[NG1] map using chimera’s fit-in-map function [32]. The initial apical monomer was then fitted into the density map manually using Coot’s transform and real-space-refine functions on individual amino acids within the model [34]. Care was taken to note the location of large amino acids (notably tryptophan, tyrosine, phenylalanine, and histidine) when manually translocating amino acids, starting from βG, which is enriched in these amino acids. DNA nucleotides were built into the map to fit the density observed, though resolution was insufficient to determine base pair identity.

Once the apical monomer and DNA were built, an asymmetric unit of 11 CP and DNA monomers was built and assigned one letter chain IDs (A, B, C, D, E, F, G, H, I, J, and K). Each monomer was independently refined manually as for the apical monomer previously and inspected for deviations from the apical monomer. For the singles, the apical chain A was used as the starting model.

A capsid comprising 110 copies of CP chains and 110 DNA chains was then built from the asymmetric unit in chimera [32]. The full capsid had manually assigned chain IDs such that the first asymmetric units were one-letter IDs previously assigned, and subsequent repeating monomers had consistent prefix IDs such that A, AA, BA, CA, DA, EA, FA, GA, HA, and IA were all equivalent to chain A. The gemini capsid was first refined using the program Phenix real-space-refinement rigid body mode, followed by the standard macro, and finally using morphing and simulated annealing [35]. Phenix real-space refinement was performed such that equivalent monomers were refined independently according to the asymmetric unit. This ensured all equivalent suffix chains (the previously described A through IA) were identical, while the different suffix chains (suffixes A and C) were non-identical.

#### 2.3.4. Structure-Based Sequence Alignment and Phylogeny

Amino acid sequence alignments were carried out with monomeric pdbs (apical chain) and primary sequences of *Geminiviridae* representing genera with no known structure using AlphaFold. Phylogeny based on this sequence alignment was determined using the phylogeny fr web service by removing gap positions [36], using PhyML to build a tree [37], and rendering the tree using TreeDyn [38]. Tree support values were determined using 100 bootstraps. Absolute and similarity identity matrices were determined from the primary sequence alignment using an in-house script. Absolute identity did not count gap positions, while for the similarity matrix, the following groups were defined as similar: aromatics—tryptophan, tyrosine, and phenylalanine; hydroxyl-containing—threonine, and serine; amide-containing—glutamine and asparagine; positive—lysine, arginine, and histidine; negative—glutamate and aspartate; hydrophobic—methionine, isoleucine, valine, leucine, alanine, and cystine; and flexible—proline and glycine. These definitions aimed to keep each residue represented only once in each group.

## 3. Results

### 3.1. Characterization and Structure Determination

The purity and capsid integrity of MSV-A[NG1] generated by agroinoculation and purified by ultracentrifugation was confirmed by silver-stained SDS PAGE and cryo-EM micrographs. The SDS gels showed two CP bands—a major band at ~30 kDa and a minor band at ~60 kDa—which is consistent with the expected molecular weight of MSV CP monomers and dimers, respectively (Appendix A). Intact single T = 1 icosahedral and gemini capsids were visualized on the vitrified micrographs (Appendix A). An approximate 20:1 ratio of the gemini (783,583) compared to the single capsids (43,111) was observed in the sample (Table 1). The 3D image reconstruction of the single and gemini capsids resulted in structures determined to 3.17 Å and 3.72 Å, respectively, using an FSC threshold of 0.143 (Appendix A). The single and gemini capsid maps displayed well-ordered side-chain density for the CP residues, and the fit of the models in the density maps was consistent with the experimentally determined resolutions (Appendix A).

### 3.2. Single-Head T = 1 Capsid Structure

#### 3.2.1. Capsid and CP–CP Interaction

Two types of single-headed capsids have been previously observed during geminivirus infections: (i) half gemini capsids (55 CPs, a capsomere absent) that package defective interfering (DI) DNA or satellite DNA [39,40], and (ii) T = 1 icosahedral that packages subgenomic ssDNA of 0.2–1.6 kb [13]. The reconstructed MSV-A[NG-1] T = 1 icosahedral has an approximate diameter of 200 Å. The protrusions on the fivefold vertices, which create a channel, are the most prominent features of the capsid, with depressions at both the two- and threefold axes (Figure 1A,B). The N-terminus of the CP is disordered, with the first ordered residue arginine 30 (R30), the main chain fitted for residues A31–S33, and the remaining residue from lysine 34 (K34) to asparagine 243 (N243) at the C-terminus showing ordered density. This is similar to several other ssDNA and RNA virus structures that have been determined, where the N-terminal residues are not resolved (reviewed in [41,42,43]). The N-termini of the geminivirus CP are highly positively charged, located in the capsid interior, and most probably interact with and counter the negative charge of the viral genome during packaging. The main secondary structure of the CP is an eight-stranded antiparallel β-barrel motif consisting of βBIDG and βCHEF, as observed in all icosahedral ssDNA structures determined to date. The β strands are separated by interconnecting loops, which are named based on the preceding and proceeding strands, for example, the HI loop connects the βH and βI strands. In addition, the structure contains two helices located within the DE and EF loops. Within the core of the capsid, βF and βG are facing the capsid lumen with the β barrel interconnecting loops facing the capsid exterior (Figure 1C). The twofold CP–CP interface is formed by the GH loop of one monomer and EF loop of the symmetry-related monomer (Figure 1D), the threefold interface is formed by the interaction of the CD loop of one monomer and the GH loop of the symmetry-related monomer (Figure 1E), and the fivefold interface is formed by the symmetry-related interactions of the DE, EF, and FG loops and βD, βG, βF, and βI strands. A complete description of these interactions is provided in Appendix A and Figure 1D–F. The two- and threefold interfaces are formed by only a few interactions (three and two H-bonds, respectively) and a minor buried surface area (320 and 220 Å^2^, respectively). In contrast, the fivefold interface has 17 H-bonds and 1120 Å^2^ buried surface area. This implies that the geminivirus fivefold interface is the most stable based on the number of hydrogen bonds and the buried surface area calculations, and points to its potential role in the initiation of capsid assembly [19,24].

#### 3.2.2. DNA-Binding Pocket

Although the T = 1 MSV-A[NG-1] capsid is non-infectious, it packages subgenomic ssDNA encoding Rep, RepA, and the LIR [13]. A cross section of the MSV-A[NG1] T = 1 capsid map reveals two regions of additional electron density not assigned to the CP (Figure 1G–I). These densities are located on the capsid interior: (i) beneath the fivefold channel, termed the fivefold plug, (ii) and at the interface between two fivefold symmetry-related CP monomers and modeled as ssDNA (Figure 1H,I). The fivefold plug blocks the pore (Figure 1G) and appears to be stabilized at the base of the pore in the capsid lumen, interacting with residues P180, K182, C181, and K197 (Figure 1I). The resolution of the density is not sufficient to determine its composition, whether it is DNA or the N-terminus of the CP or both, whereas the DNA-binding pocket was unambiguously modeled as nine nucleotides and interacts with adjacent monomers involved in the intra-fivefold CP interactions (Figure 1H,I). The consensus sequence of the fitted nucleotides RRRRYYYYR hairpin (Y represents pyrimidines (cytosine and thymine), and R represents purines (adenine and guanine)). The fitted nucleotides and the corresponding interacting residues of the CP are listed in Appendix A. The proximal portion of the nucleotide chain interacts with one CP, folds back on itself at the fifth nucleotide, and the distal portion of the nucleotide chain interacts with an adjacent fivefold symmetry CP (Figure 1H,I). These interactions include π stacks and hydrogen and hydrophobic bonds, and appear to be important in further stabilizing the pentameric unit, which has been proposed to be crucial to the assembly of the capsid (Appendix A) [19,24].

### 3.3. Gemini Capsid Structure

#### 3.3.1. Capsid and CP–CP Interactions

The MSV-A[NG1] overall capsid architecture is consistent with previously reported geminivirus structures [17,19,44]. The capsid dimension is ~240 × 390 Å and is composed of two pseudo-T = 1 icosahedral heads (hemi-capsid), joined at the waist or equatorial region with a 20° twist of the two heads in the longitudinal axis of the capsid (Figure 2A,B). Each head of the gemini capsid is assembled from three distinct building blocks or pentamers, which are located at the peak (apical—A), shoulder (peripentonal—P) and waist (equatorial—E) of the capsid (Figure 2A). Each pentamer, regardless of its position on the capsid, projects radially outward, while the two- and threefold axes occupy shallow depressions on the capsid surface (Figure 2A). The arrangement of the monomers of the apical pentamers more closely resembles the pentamers of the T = 1 icosahedral capsid, with increasing deviation from icosahedral symmetry towards the equatorial region (Figure 2A). The capsid is composed of 110 CP monomers, equivalent to two T = 1 capsids less than a pentamer from each at the equatorial interface (Figure 2B). The geminivirus capsid viral asymmetric unit is composed of 11 CPs, labeled monomers A–K, which are multiplied 10× to give the 110 CP monomers (Figure 2C). The assignment of the asymmetric unit in the lower portion of the gemini capsid is shown in Figure 2C, labeled monomers A^1^–K^1^. Monomer A, located at the apical region, has the same orientation as the T = 1 icosahedral unit, monomers B–F compose the pentameric unit located at the capsid peripentonal region, and monomers G–K form the pentameric unit at the equatorial region of the capsid (Figure 2A,C).

The secondary structure of the CP monomer is identical to that of the T = 1 capsid (Appendix A). Superposition of monomer A onto C, I, and H yields C with RMSD of 0.4, 0.2, and 0.6 Å, respectively, and the N-termini exhibiting the most structural variability (Appendix A). This is consistent with the observed variability in the N-terminal lengths of these equivalent monomers in AYVV [19]. The N-termini of chain H and chain I were predicted to extend across the equatorial interface, and this interaction was proposed to be critical for the formation of the gemini capsid. This, however, is not observed in the MSV-A[NG1] capsid.

Similarly, the number of interactions and buried surface area of the two-, three-, and fivefold interfaces of the T = 1 icosahedral and each hemi-head of the MSV-A[NG-1] gemini capsid have similar values, as illustrated in Figure 2A,B and Appendix A. There is, however, an increase in the number of H-bonds and buried surface area between the monomers at the equatorial interface, as monomer H and I rotate almost 90° radially outward to accommodate monomer H^1^ and I^1^ across the equatorial interface and specifically at α-helix B, located in the EF loop (Figure 2D,E and Appendix A). Radial movement of the CP at the equatorial interface was also shown in the ACMV structure and illustrates the unique conformations and interactions of the CP of the geminivirus capsid at the equatorial interface to accommodate the viral genome, which is required to assemble this unique capsid architecture.

#### 3.3.2. CP–Genome Interactions

The gemini capsid packages a genome of ~2.5 kb, and the nucleotide-binding pocket described above for the single T = 1 capsid is conserved in both MSV-A[NG1], ACMV and AYVV gemini capsids, positioned at the base of the interface between fivefold related CPs [19,44]. Similar to the structure of the T = 1 capsid, a cross-sectional view of the nucleotide density on the gemini capsid interior shows a pentameric DNA scaffold or cage (Figure 3A). There is no ordered DNA density at the twofold, threefold, or equatorial interface. The DNA density also appears to be in close proximity to the base of the fivefold plug. The resolution of the gemini capsid map ranges from 3.2 to 4.6 Å. The density of the CP is more ordered, with lower local resolution for the DNA plug and the polynucleotide (Appendix A). The fit of the nucleotide density is shown in Figure 3B. The resolution is only sufficient to determine whether the nucleotide is a purine or pyrimidine. Each polynucleotide interacts with two adjacent fivefold related CP monomers, that is, each CP binds 2 × 9 mers (polynucleotides) (Figure 3C–F). Eight of the 10 residues that interact with the 9-mer nucleotides adjacent to the N-terminus of the CP are basic, with the remaining two aromatic and one hydrophobic (Figure 3C–F and Appendix A).

The basic residues stabilize the phosphate backbone of the 9 mer, and the aromatic residues allow for π-stacking with the nucleotide base (Figure 3D). The total list of CP residues that interact with the 9mer are listed in Appendix A. The bases also π-stack with each other to form a loop. The observed conserved nucleotides and nucleotide-binding pocket in both MSV-N[A] and AYVV implied that the ACMV may also exhibit this feature, although there was no nucleotide density fitted in the ACMV density. The superposition of the ACMV and AYVV maps show similar unassigned density in the ACMV map, which was fitted with the AYVV nucleotides. This observation and the density seen in the MSV-N(A) map strongly imply that the nucleotide-binding pocket is a conserved feature of *Geminiviridae*.

#### 3.3.3. Conservation of Geminivirus Structure

An unrooted phylogenic tree using representative member sequences from all the 14 geminivirus genera demonstrated multiple diverse branching, as expected, due to the differences in amino acid sequences among the genera (Figure 4A, Appendix A). Comparative sequence and structural analysis of the CP of the available structures of MSV-N[A], AYVV, ACMV and two additional species, TPCTV (Topovirus) and CCDAV (Citildovirus), not surprisingly demonstrated the conservation of the core of the capsid. There were, however, distant nodes observed in the tree, specifically TALCV and TPCTV. They formed distant branches of the phylogenic tree when compared to MCLV and CCDAV (Figure 4A), and they had a direct correlation with the less conserved sequences observed in the interconnecting loops (Figure 4B and Figure 5A,B). Based on the superposition of the five CP monomers (MSV, ACMV, AYVV, TPCTV, and CCDVA), seven structurally different regions were identified and defined as variable regions (VR I–VII), based on the distance of three consecutive residues with Cα positions ≥3 Å from residues equivalent to the MSV coordinates (Figure 5A–F).

The most distinct variabilities observed on the capsid surface are localized at the fivefold vertex radially protruding outward, VR VI localized around the threefold axis, and VR IV located on the wall of the twofold depression (Figure 5A–F). The only variable region that is not surface-accessible is VR I, which resides in the lumen of the capsid and has been shown to be important for ssDNA binding and encapsidation. There is low amino acid sequence identity between the different geminivirus genera ranging from 20% to 30%, but relatively high structural similarity ranging from 60% to 95% within the genus (Appendix A). This observation supports the theory that the β-barrel motifs are determined by a small, conserved percentage of the CP sequences, with greater variation in the connecting loop to facilitate adaptation to the host vector.

## 4. Discussion

The structure of the MSV-A [NG1] single capsid is the first T = 1 geminivirus capsid determined (Figure 1). The main distinction between the gemini and the T = 1 capsid is the length of the genome encapsulated [13,24] and the CP–CP interactions at the equatorial interface observed in the gemini capsid (Figure 2). The isolation and identification of a triple-head gemini capsid assembled in the presence of a larger genome [40] and the structural characterization of the gemini and single MSV-A [NG1] capsid point to a virus that has emerged to assemble the twinned or triple-head architecture relative to the increasing size of the available genome illustrated in Figure 6. A cross-sectional view of the modeled nucleotide in the gemini capsid shows a DNA scaffold, which potentially forms first and may lay the foundation for the assembly of the capsid (Figure 3A).

The high degree of interdigitation of the CP of the equatorial interface implies that geminivirus capsid assembly is initiated at the equatorial interface during genome replication and viral mini-chromosome formation. The viral mini-chromosome persists extra-chromosomally and is a template for transcription [45,46], and the newly formed ssDNA during replication may potentially serve the same role in capsid assembly. This step is potentially followed by the addition of the pentamers to the peripentonal and apical region of the genome scaffold (Appendix A). This model is consistent with the tiling approach to virus capsid assembly, for non-equivalent subunit arrangement described for quasi-equivalent virus assembly observed in MS2, and hepatitis B [47]. It is unlikely that the nonanucleotide sequence observed in the MSV structure is the conserved stem loop observed in all geminiviruses, as this only occurs once in the genome [21,45,48] and is not repeated 110 times. It should be noted that empty geminiviruses have not been observed from a wild-type infection. This would imply that the folded genome may be the critical first step in the geminivirus capsid assembly, as previously proposed for AYVV [19].

The most predominant structural feature of the capsid is the pentamers/capsomers that extend radially outward from the capsid. This is also the most surface-accessible feature for host and vector interactions. The pores formed at the center of the pentamers provide an entryway to the capsid interior. The fivefold pore is most likely utilized for the externalization of the viral genome or the site for the externalization of the N-termini of the CP, which is a common feature observed in other ssDNA viruses, for example, in Parvoviruses, such as adenoassociated virus (AAV), canine parvovirus (CPV) and minute virus of mice (MVM) [49,50,51]. Compared to other viruses that exhibit density in their fivefold pores, the geminiviruses do not have any corresponding density. There is, however, unmodeled density under the pores, which interacts with K179, P180, C181, and K182. The PCK (residue 180–182) motif is conserved in all Mastrevirus, located in the FG loop, which is involved in extensive fivefold interactions [52]. Geminivirus CP mutational analysis, and CP–CP binding studies [53] reveal the importance of the fivefold interactions to capsid assembly. The purification and characterization of pentameric CP subassemblies from agroinoculated wild-type geminivirus infection [24] and buried surface area information derived from the available cryo-EM structures all show the utilization of a pentameric CP intermediate in the assembly of the capsid. In addition, the CP–CP and CP–genome interactions are critical for the assembly and transmission of the geminivirus. The nucleotide modeled accounts for ~30% of the total genome and the fitted CP ~88% of the protein. The CP is multifunctional and critical to the assembly and transmission of this highly pathogenic virus. The structural and functional annotation of the CP in the assembled virus provides information on the regions of the virus capsid that can be targeted to inhibit infection and transmission.

It has been proposed that one of the major driving forces of genetic variability and geographic distribution in geminivirus is the high degree of recombination events, especially for Begomovirus [54]. This information prompts a fundamental question: What residues contribute to the conserved secondary structure responsible for the core capsid formation, and is there a specific role of the interconnecting VRs? To date, there has been one subatomic and one high-resolution geminivirus structure determined, both from the *Begomovirus* genus [19,44]. The addition of the MSV-A [N] structure from the *Mastrevirus* genus and the use of the predictive capabilities of AlphaFold provides an opportunity to compare the CP sequence and structures of representative members of different genera of the geminivirus family. Superposition of the CP monomers of five representative members from four different genera and five distinct species was used to define the conserved secondary elements. These are colored light gray and represent the conserved capsid core responsible for the assembly and the stability of the capsid (Figure 5). There is significant amino acid conservation between the CPs of viruses from the same or similar genera, which would imply that the CP may play a role in virus vector interaction. For example, there is ~85% structural similarity between the CP of AYVV and ACMV and ~60–70% between MSV and ACMV and AYVV (Appendix A). VRs surrounding the two- and threefold depression and the fivefold protrusion on the capsid exterior (Figure 5C–F) are most likely responsible for geminivirus life-cycle differences and vector preferences. Mutational analysis in the VRs of the CP of three Begomoviridae, Abutilion mosaic virus (AbMV), and Sida golden mosaic virus (SgMV) from Honduras and Costa Rica identified amino acids 123–174 to be critical for vector (*Bemisia tabaci*) transmission. These residues form a part of VRII, VRIII, and VRIV located at the twofold depression and the base and wall of the fivefold protrusion of the virus capsid (Figure 5) [55]. Other begomoviruses investigated also utilize these VRs in the *Geminiviridae* to confer genus-vector specificity [53,56], for example, a variant of watermelon chlorotic stunt virus with a mutation in residue N131D (conserved in the Begomovirus) located in VRII also oblates transmission by the vector Bemisia tabaci [56]. These variable regions offer a toolbox of potential sites on each geminivirus capsid that can be targeted by a specific antibody (administered to the plant) that will prevent binding of the virus to a specific vector. The available sequence and structure alignment provide information on potential sites on the geminivirus capsid that can be mutated or modified to evaluate their importance in vector transmission.

In summary, the available structures of MSV-A [NG1] T = 1 and gemini capsids provide a detailed view of the Mastrevirus capsid. These structures provide insight into the CP–CP and CP–genome interactions in both capsids, specifically the role of the genome on capsid assembly and multimerization. Finally, the phylogenetic analysis of the CP from each genus provides a qualitative understanding of the CP variations between them. This, combined with the only available high-resolution structures from 2 of the 14 genera, was used to visually annotate sites on the virus capsid that may be important for capsid assembly, stability, and vector transmission. These identified sites can be potential targets for the design of antiviral therapies.

## Figures and Tables

**Figure 1 viruses-16-01861-f001:**
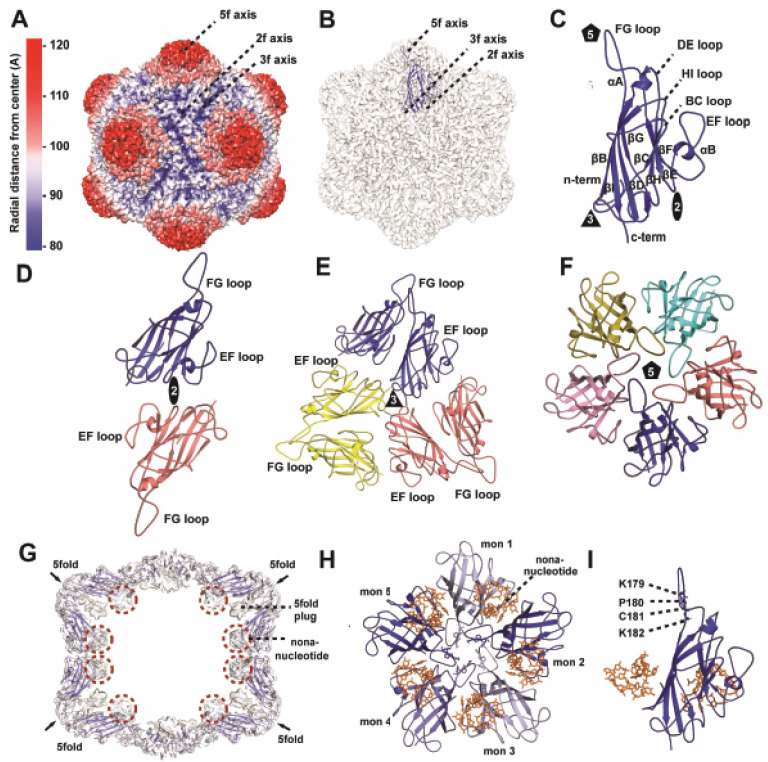
MSV-A [NG1] T = 1 structure and fitted model. (**A**) Surface representation radially colored according to the color key. (**B**) Surface colored gray and fitted with a monomer that is colored blue, labeled at the two-, three-, and fivefold axes. (**C**) Cartoon representation of the fitted monomer, which is colored blue. The secondary structural elements are labeled βBIDG, βCHEF, αA, and αB. Cartoon representation of the (**D**) dimer, (**E**) trimer, and (**F**) pentamer. The icosahedral twofold, threefold, and fivefold axes are represented as an oblong, triangle, and pentagon, respectively. (**G**) Orthogonal view of the capsid map fitted with the 60-mer model, showing the fivefold plug, DNA binding pocket under the fivefold pore and the base of the 5-pentamer. (**H**) 5CP monomer and nucleotide. (**I**) Monomer with interacting nucleotide and K179, P180, C181, and K182.

**Figure 2 viruses-16-01861-f002:**
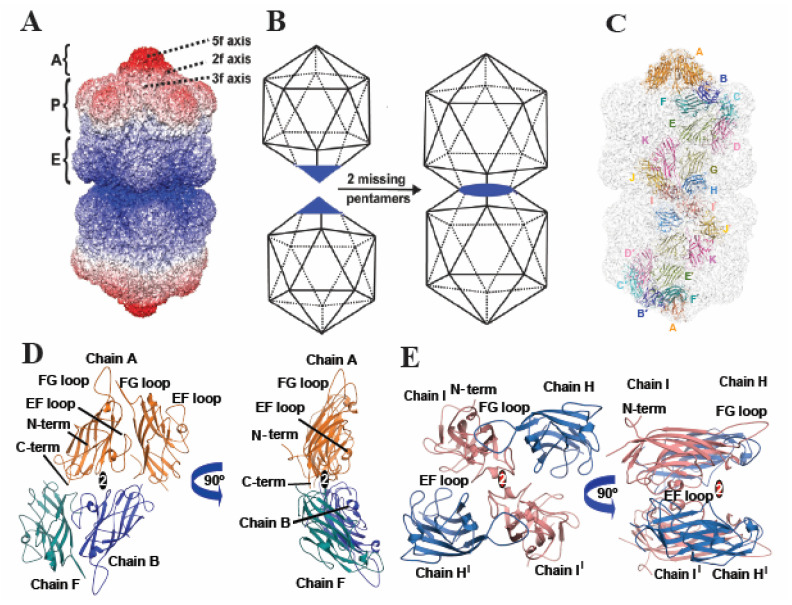
MSV-A[NG1] gemini capsid structure and fitted model. (**A**) Surface representation radially colored, the equatorial region is colored blue, and the apical point is colored red. (**B**) Schematic of two T = 1 capsid (60 monomers each), each showing a pentamer colored blue. If two pentamers (10 CP monomers) are removed, then the resulting structure would form a double head (110 mer) similar to the gemini capsid. (**C**) Surface representation colored gray and fitted with the viral asymmetric unit labeled A–K and its duplicate in the lower head labeled A′–K′, the viral asymmetric unit with the twofold, threefold, and fivefold axes represented as an oblong, triangle, and pentagon, respectively. (**D**) View of the twofold interaction between the apical and peripentonal monomers (A:A and B:F). (**E**) View of the twofold interaction between the peripentonal and equatorial monomers (H–I and H′–I′). The icosahedral twofold axis is represented by an oval that is colored black and the equatorial twofold axis is colored red.

**Figure 3 viruses-16-01861-f003:**
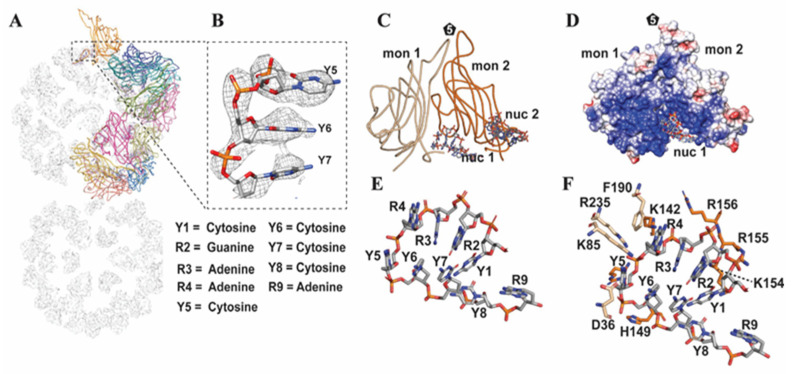
MSV-A [NG1] gemini capsid and genome. (**A**) Viral asymmetric unit on a scaffold generated from the modeled DNA density. (**B**) View of 6 of the nucleotides, labeled R1–Y4. (**C**) Ribbon diagram of the two of the monomers from a pentamer and interacting nucleotides. (**D**) Charged surface representation of the monomers in (**C**) with the nucleotides shown as a stick diagram. (**E**) Model of the nucleotide with the purines labeled R and pyrimidines labeled Y. (**F**) Model of the nucleotide and the interacting residues from monomer 1 (light brown) and monomer 2 (orange).

**Figure 4 viruses-16-01861-f004:**
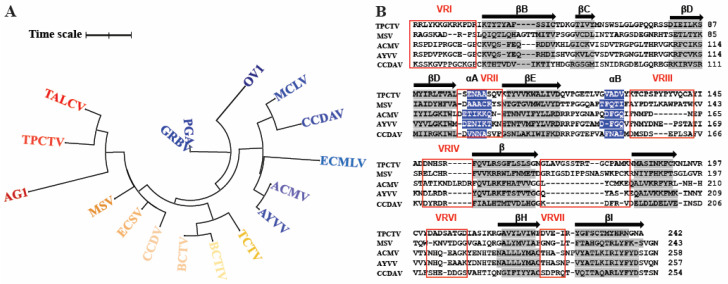
Phylogenic and structure-based sequence alignment of representative geminivirus. (**A**) Phylogenic analysis of representative members of all the geminivirus genera. The time scale shows substitution distance at each stroke as 1. (**B**) Sequence alignment of TPCTV, MSV, ACMV, AYVV, and CCDAV.

**Figure 5 viruses-16-01861-f005:**
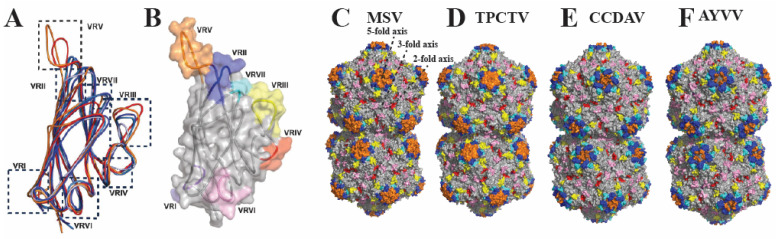
Comparison of geminivirus capsid/CP structures. (**A**) Superposition of the apical monomers of TPCTV, MSV, ACMV, AYVV, and CCDAV and VRs assigned based on RMSD of the mainchain ≥ 3 Å. (**B**) Surface representation of the monomer of MSV with VRI-VII (**C**) MSV, (**D**) TPCTV, (**E**) CCDAV, and (**F**) AYVV gemini capsids, colored based on the location of the VRs.

**Figure 6 viruses-16-01861-f006:**
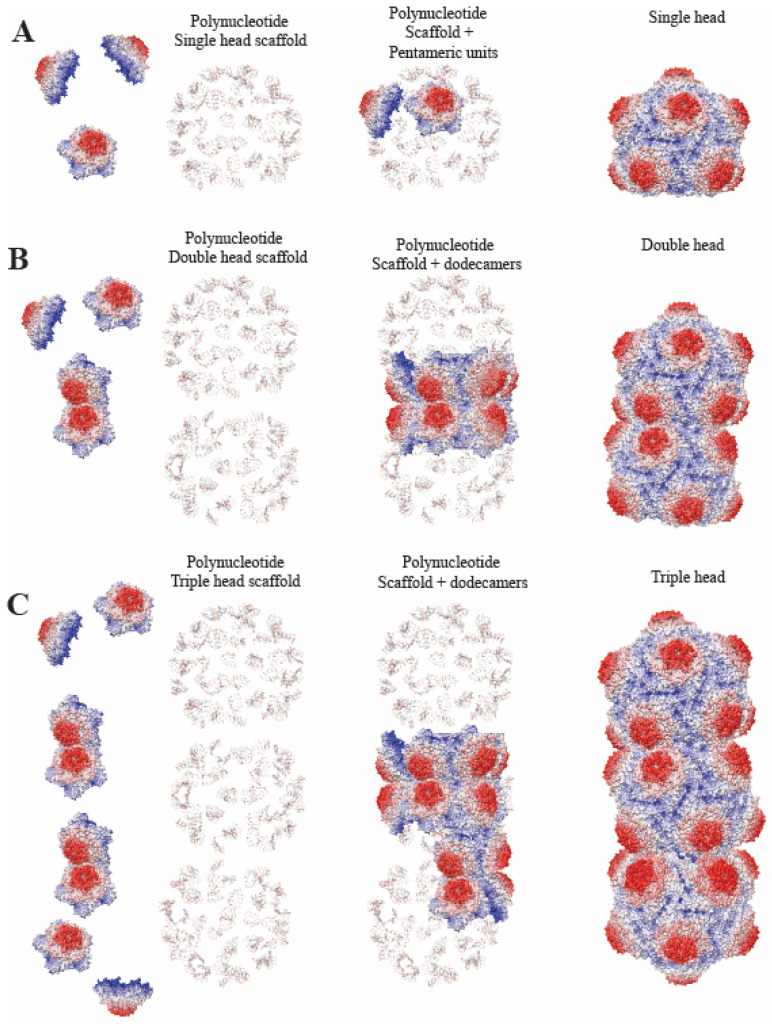
Model of geminivirus capsid assembly utilizing polynucleotide oligomerization. (**A**) The polynucleotide scaffold of MSV-A[N] of fewer than five polynucleotide chains and equivalent pentameric unit assemble to form the single T = 1 capsid. (**B**) Polynucleotide double-head scaffold with pentamers and dimers of pentamers (dodecamers). Dodecamers assemble the equatorial interface first, then the other pentamers are added to the peripentonal and then the apical region. (**C**) Polynucleotide triple-head scaffold with pentamers and dimers of pentamers (dodecamers) sequentially added to assemble the triple head capsid.

**Table 1 viruses-16-01861-t001:** Summary of cryo-EM data collection, reconstruction, and refinement statistics.

Single-Particle Reconstruction Parameters		
	Gemini Capsid	Single Capsid
PDB ID	8UGQ	8UH4
EMDB ID	42,232	42,246
Total number of micrographs	1408	
Defocus (µm)	3.627–0.947	
Electron dose (e Å^−2^ frame^−1^)	2.04	
Frames per micrograph	29	
Pixel size (Å pixel^−1^)	1.01	
Particles	783,583	43,111
Resolution (Å) FSC_0.143_	3.17	3.72
Symmetry imposed	D5	I
**Model refinement statistics**		
Asymmetric units		
CP monomers	11	1
DNA	11	1
Map CC	0.8169	0.8097
Map-Model-FSC_0_	3.27	3.8
FSC_0.143_	3.39	3.9
FSC_0.5_	3.70	4.2
RMSD bond (Å)	0.003	0.006
RMSD angle (°)	0.536	0.834
All-atom clash score	4.11	5.25
Ramachandran% Favored	97.8	97.5
Ramachandran% Allowed	2.2	2.5
Ramachandran% Unfavored	0.0	0.0
Rotamer outliers (%)	0.0	0.0
C-β deviation (%)	0.0	0.0

## Data Availability

The MSV-A[NG1] cryo-EM reconstructed density maps and models were deposited in the Electron Microscopy Data Bank (EMDB; https://www.ebi.ac.uk/emdb/, accessed on 22 November 2024) with accession numbers of 42246 and 42232) and the Protein Data Bank with accession numbers 8UH4 and 8UGQ for the single and gemini capsids, respectively.

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
