# Peer review of "Maize Streak Virus: Single and Gemini Capsid Architecture"

_viruses, 2024, doi:10.3390/v16121861_

Round 1
Reviewer 1 Report
Comments and Suggestions for Authors
Excellent paper! Well written and of great interest. A few references which may be useful:
Guevara-Rivera EA, Rodríguez-Negrete EA, Lozano-Durán R, Bejarano ER, Torres-Calderón AM, Arce-Leal ÁP, Leyva-López NE, Méndez-Lozano J. From Metagenomics to Ecogenomics: NGS-Based Approaches for Discovery of New Circular DNA Single-Stranded Viral Species. Methods Mol Biol. 2024;2732:103-117. doi: 10.1007/978-1-0716-3515-5_7. PMID: 38060120.
Morais IJ, Inoue-Nagata AK, Nakasu EYT. Construction of Geminivirus Infectious Clones for Agroinoculation into Plants. Methods Mol Biol. 2024;2724:47-64. doi: 10.1007/978-1-0716-3485-1_4. PMID: 37987897.
Would this paper enable the development of pseudovirions of geminiviruses with different genomes, or other types of nucleic acid? Could this information be used to make genome edited plants? Just wondering about the potential of the capsid as a delivery vehicle.
Author Response
Reviewer 1:
Excellent paper! Well written and of great interest. A few references which may be useful:
- Guevara-Rivera EA, Rodríguez-Negrete EA, Lozano-Durán R, Bejarano ER, Torres-Calderón AM, Arce-Leal ÁP, Leyva-López NE, Méndez-Lozano J. From Metagenomics to Ecogenomics: NGS-Based Approaches for Discovery of New Circular DNA Single-Stranded Viral Species. Methods Mol Biol. 2024;2732:103-117. doi: 10.1007/978-1-0716-3515-5_7. PMID: 38060120.
- Morais IJ, Inoue-Nagata AK, Nakasu EYT. Construction of Geminivirus Infectious Clones for Agroinoculation into Plants. Methods Mol Biol. 2024;2724:47-64. doi: 10.1007/978-1-0716-3485-1_4. PMID: 37987897.
Would this paper enable the development of pseudovirions of geminiviruses with different genomes, or other types of nucleic acid? Could this information be used to make genome edited plants? Just wondering about the potential of the capsid as a delivery vehicle.
Response: We thank the reviewer for the suggested references, and we have modified the document to include the references. In response to your question, it is an intriguing thought. The information provided in the manuscript can potentially be used to develop geminiviruses with mixed CP sequences (from different genera) or mixing different elements of the genome to generate a pseudo-geminivirus, and definitely has potential as a delivery vehicle.
Reviewer 2 Report
Comments and Suggestions for Authors
The manuscript "Maize Streak Virus: Single and Gemini Capsid Architechture" by Bennett et al, deals with the structure of single and gemini capsids for Maize streak virus, using a variety of methods, including cryo-EM and AlphaFold analysis.
The manuscript is well written and very illustrative with figures that guide the reader to understand the structure/function relationships of the CP monomers from the virus. Since there are only a few of the solved CPs available for these family of viruses, the manuscript is very relevant.
We have a few comments on the manuscript that may help improving it.
Figure 2. "the equatorial regio is colored", it should read "region".
Lane 404, "different geminivirus genres", it should read "genera".
Lane 482, "to the conserve secondary structure", it should read "conserved"
Lane 487, "of different genre of the", it should read "genera"
Lane 490, "secondary elements these are", it should read "secondary elements. These are.."
Lane 500, "Bemisia tabacia", it should read "Bemisia tabaci"
Lane 505," Bemicia tabacia", it should read Bemisia tabaci, in italics.
Lane 507, the authors mention that the geminivirus capsid can be targeted by a specific antibody to prevent binding to a vector, however, plants don't produce naturally antibodies, maybe the authors should clarify that this antibody should be engineered into the plants to be able to stop the transmission?
Author Response
Reviewer 2:
The manuscript "Maize Streak Virus: Single and Gemini Capsid Architechture" by Bennett et al, deals with the structure of single and gemini capsids for Maize streak virus, using a variety of methods, including cryo-EM and AlphaFold analysis.
The manuscript is well written and very illustrative with figures that guide the reader to understand the structure/function relationships of the CP monomers from the virus. Since there are only a few of the solved CPs available for these family of viruses, the manuscript is very relevant.
We have a few comments on the manuscript that may help improving it.
Figure 2. "the equatorial regio is colored", it should read "region".
Lane 404, "different geminivirus genres", it should read "genera".
Lane 482, "to the conserve secondary structure", it should read "conserved"
Lane 487, "of different genre of the", it should read "genera"
Lane 490, "secondary elements these are", it should read "secondary elements. These are.."
Lane 500, "Bemisia tabacia", it should read "Bemisia tabaci"
Lane 505," Bemicia tabacia", it should read Bemisia tabaci, in italics.
Lane 507, the authors mention that the geminivirus capsid can be targeted by a specific antibody to prevent binding to a vector, however, plants don't produce naturally antibodies, maybe the authors should clarify that this antibody should be engineered into the plants to be able to stop the transmission?
Response: We thank the reviewer for the comments, and we have modified the document in the same lines listed in the comment, according to your suggestions.
Reviewer 3 Report
Comments and Suggestions for Authors
Bennett et al. present the cryo-EM structures of the two capsid forms of the Maize Streak Virus, that are icosahedral single and twinned capsid. Authors went on analyzing these structures in detail to reveal the CP-CP and CP-DNA interactions. Furthermore, comparative structural and phylogenetic analysis is conducted for other representative members of the Geminiviridae. Overall I believe these results are important but I am concerned (detailed below) about the structure of DNA fragment and associated CP-DNA interactions described in the manuscript which should be addressed before the acceptance of the manuscript. My other concerns are also listed below.
#The CP is multifunctional, it accumulates in the nucleus, interacts with the genomic DNA, transports the DNA from the nucleus to the cytoplasm, and is required for the encapsulation of ssDNA, insect transmission, and systemic infection [14, 15].
It this sentence it should be explicitly mentioned that CP forms the virus capsid which performs the listed tasks.
#Currently, there is one near atomic and a high resolution Geminiviridae, Begomovirus genus structures determined to date, these are African cassava mosaic virus (ACMV) (PDB ID:6EK5) and Ageratum yellow vein virus (AYVV) (PDBID:6F2S) to a resolution of 4.2 Å and 3.3 Å, respectively [18, 19].
The first part of this sentence contradicts the latter part. There are more than one structures available.
I have checked the map and model of Gemini capsid solved in this study. While the density for CP is nicely resolved, the density for DNA fragment is not convincing. See the image below with the DNA fragment shown in red. Could authors clarify this? With this issue, all the CP-DNA interactions described are in question.
Similarly, the density for CP residues Ala31-Ser33 is not convincing too. Please justify.
Several important cryo-EM data processing and model building metrics are missing as listed below. Addition of these metrics would be nice for readers to have cleared picture about cryoEM data processing.
· Map-to-model FSC is missing, should be shown in Table 1
· The local resolution map is missing, Should be shown at least in supplementary material
· Overall workflow for cryoEM data processing can shown as additional supplementary figure
Author Response
Bennett et al. present the cryo-EM structures of the two capsid forms of the Maize Streak Virus, that are icosahedral single and twinned capsid. Authors went on analyzing these structures in detail to reveal the CP-CP and CP-DNA interactions. Furthermore, comparative structural and phylogenetic analysis is conducted for other representative members of the Geminiviridae. Overall, I believe these results are important but I am concerned (detailed below) about the structure of DNA fragment and associated CP-DNA interactions described in the manuscript which should be addressed before the acceptance of the manuscript. My other concerns are also listed below.
Response: We thank the reviewer for the comments, and we have modified the document according to your suggestions.
#The CP is multifunctional, it accumulates in the nucleus, interacts with the genomic DNA, transports the DNA from the nucleus to the cytoplasm, and is required for the encapsulation of ssDNA, insect transmission, and systemic infection [14, 15].
It this sentence it should be explicitly mentioned that CP forms the virus capsid which performs the listed tasks.
Response:
We added a statement in line 57 to explicitly state that the CP assembles the geminivirus capsid. The other functions of the CP are not assumed to be related to the assembled virus capsid, the CP monomer has been previously shown to interact with both ss, dsDNA and the movement protein (MP) etc.
#Currently, there is one near atomic and a high resolution Geminiviridae, Begomovirus genus structures determined to date, these are African cassava mosaic virus (ACMV) (PDB ID:6EK5) and Ageratum yellow vein virus (AYVV) (PDBID:6F2S) to a resolution of 4.2 Å and 3.3 Å, respectively [18, 19].
The first part of this sentence contradicts the latter part. There are more than one structures available.
Response:
We have modified the statement in lines 61 and 62 to read, ‘Currently, two Geminiviridae, Begomovirus genus structures were determined to date, these are African cassava mosaic virus (ACMV) (PDB ID:6EK5) and Ageratum yellow vein virus (AYVV) (PDB ID:6F2S) to a resolution of 4.2 Å and 3.3 Å, respectively [18, 19]’
I have checked the map and model of Gemini capsid solved in this study. While the density for CP is nicely resolved, the density for DNA fragment is not convincing. See the image below with the DNA fragment shown in red. Could authors clarify this? With this issue, all the CP-DNA interactions described are in question. Similarly, the density for CP residues Ala31-Ser33 is not convincing too. Please justify.
Response:
The addition of the local resolution map, does illustrate that the density of the nucleotide is lower than that of the CP. However, the density of the DNA is still sufficient to confirm whether the polynucleotide is a pyrimidine or a purine. The image below has the sigma level of the map at 6.64 which is a higher value than is required to accurately fit the model into the map. We have modified the document to more accurately describe the difference in the local resolution.
Several important cryo-EM data processing and model building metrics are missing as listed below. Addition of these metrics would be nice for readers to have cleared picture about cryoEM data processing.
- Map-to-model FSC is missing, should be shown in Table.
Response: The map to model FSC was added to Table 1
- The local resolution map is missing, Should be shown at least in supplementary material
Response: The local resolution map was added to the Supplementary data as Figure S3 and
the document modified to include an explanation of the figure.
Overall workflow for cryoEM data processing can be shown as additional supplementary figure.
Response: This was added to the supplemental data as Figure S4
Round 2
Reviewer 3 Report
Comments and Suggestions for Authors Authors have replied to most of my comments but the following three comments need to be addressed further. 1. Authors argue that the density for DNA is visible at lower threshold level which is good enough whether the polynucleotide is a pyrimidine or a purin. If so then the DNA with the density should be shown as a main figure in the manuscript as the CP-DNA interaction is the important part of the manuscript.
